# Heterogeneous Cross-Coupling over Gold Nanoclusters

**DOI:** 10.3390/nano9060838

**Published:** 2019-06-01

**Authors:** Quanquan Shi, Zhaoxian Qin, Hui Xu, Gao Li

**Affiliations:** 1College of Science, Inner Mongolia Agricultural University, Hohhot 010018, China; qqshi@dicp.ac.cn (Q.S.); yqfxxuhui@163.com (H.X.); 2State Key Laboratory of Catalysis, Dalian Institute of Chemical Physics, Chinese Academy of Sciences, Dalian 116023, China; qinzhaoxian@dicp.ac.cn

**Keywords:** gold nanocluster, cross-coupling, Ullmann hetero-coupling, Sonogashira coupling, Suzuki coupling, A^3^−coupling, catalytic mechanism, ligand removal

## Abstract

Au clusters with the precise numbers of gold atoms, a novel nanogold material, have recently attracted increasing interest in the nanoscience because of very unique and unexpected properties. The unique interaction and electron transfer between gold clusters and reactants make the clusters promising catalysts during organic transformations. The Au*_n_*L*_m_* nanoclusters (where L represents organic ligands and *n* and *m* mean the number of gold atoms and ligands, respectively) have been well investigated and developed for selective oxidation, hydrogenation, photo-catalysis, and so on. These gold clusters possess unique frameworks, providing insights into the catalytic processes and an excellent arena to correlate the atomic frameworks with their intrinsic catalytic properties and to further investigate the tentative reaction mechanisms. This review comprehensively summarizes the very latest advances in the catalytic applications of the Au nanoclusters for the C−C cross-coupling reactions, e.g., Ullmann, Sonogashira, Suzuki cross-couplings, and A^3^−coupling reactions. It is found that the proposed catalytically active sites are associated with the exposure of gold atoms on the surface of the metal core when partial capping organic ligands are selectively detached under the reaction conditions. Finally, the tentative catalytic mechanisms over the ligand-capped Au nanoclusters and the relationship of structure and catalytic performances at the atomic level using computational methods are explored in detail.

## 1. Introduction

Since the work of Haruta’s group in the late 1980s [1], supported gold nanoparticles with a particle size in the range of 3–20 nm have played a central role in a variety of reactions such as selective oxidation [2,3,4,5], hydrogenation [6,7,8], and photocatalysis [9,10]. These conventional gold catalysts have been realized via deposition–precipitation and co-precipitation impregnation with oxides by controlling the pH of the synthetic system. These obtained gold nanoparticles are usually polydisperse, which is a major issue in fundamental catalysis and investigations [11]. For example, the size-hierarchy of Au nanoparticles are often averaged out in polydispersion. It is difficult to correlate the relationship between the catalytic properties and the structure of the nanoparticles. Therefore, developing nanostructured catalysts with specific morphology (e.g., nanosheet, nanocube, and nanorod) is highly desirable to overcome this issue, further promoting the development of crystal-identified model catalysts [12,13].

On the other hand, the remarkable developments in the synthesis of atomically precise gold nanoclusters have been achieved in recent decades, which opens a new burgeoning area in nanoscience [14,15]. These gold nanoclusters are comprised of a few dozen to a few hundred gold atoms and protecting organic ligands (e.g., thiolate, phosphine, and alkyne), and the size is ultra-small, usually ca. 0.6–2 nm. Some gold nanoclusters are identifiable by X-ray crystallography technology. It offers a big opportunity for in-depth understanding of the relationship of the catalytic properties and the active-site structure at the atomic level [16].

It is observed that these gold nanoclusters exhibit good catalytic performance in heterogeneous catalysis (e.g., selective oxidation and hydrogenation) [16,17,18]. Sometime, the gold clusters show better catalytic behavior (e.g., activity and product selectivity) than the corresponding Au nanoparticles, because of their high surface-to-volume ratio (reaches up to ~100%), surface geometric effect (e.g., low-coordinated Au^δ+^ atoms, 0 < δ < 1), and the unique electronic properties and the quantum size effect. Furthermore, the protecting organic ligands can improve the product selectivity due to their electronic factors and steric hindrance and weak interaction (e.g., π-π interaction) between the reactants and cluster surface ligands during the catalysis process. In the recent decade, these gold nanoclusters exhibited good catalytic activity in the cross C–C couplings, e.g., Ullmann hetero-coupling, Suzuki and Sonogashira coupling, and A^3^-coupling [19,20]. Traditionally, these catalyzed C–C coupling reactions are over the Cu, Pd, and Pt complexes and particles in the previous literatures [21,22,23].

In this review, we aim to provide an overview focused on the Au nanocluster-catalyzed coupling reactions, e.g., Ullmann hetero-coupling of Ar-I, Suzuki cross-coupling of PhB(OH)_2_ and Ph-I (IB), Sonogashira cross-coupling of IB and Ph–C≡C–H (PA), and A^3^-coupling (Scheme 1). The pathway of the selective detachment of the surface protecting ligands (under the reaction conditions), giving the catalytically active sites, is well discussed. Moreover, the proposed reaction pathway and mechanisms of these carbon–carbon coupling reactions were thoroughly summarized based on the precise framework of gold nanoclusters (e.g., two Au_25_) as the mode theoretical calculations.

## 2. Activation of Ph–*B*(*OH*)_2_, Ph–*I*, and C≡C–*H* Bonds over Au: Theoretical Simulation

The activation of the Ph–*B*(*OH*)_2_, Ph–*I*, and C≡C–*H* bonds over the gold clusters plays an important role and step in the cross-coupling reactions, e.g., Ullmann hetero-coupling, Sonogashira coupling, Suzuki coupling, and A^3^−coupling. The adsorption and activation process of the reactants over the different well-defined gold facets and sites are distinct [24,25,26,27]. The density functional theory (DFT) calculations should be a feasible and fast method to explore the activation of Ph–*B*(*OH*)_2_, C–*I*, and C≡C–*H* bond by gold nanoclusters.

Firstly, the shape controlled Au nanoparticles (e.g., Au nanorod) with well-defined surfaces and morphologies are well investigated [28,29], providing a platform to establish site–activity relationships and to pursue the understanding of heterogeneous processes. Therefore, the gold nanorod is chosen for the theoretical simulation of the activation of Ph–*I* and C≡C–*H* bonds. DFT studies showed that the Ph-*I* reactant adsorbs onto the Au(100) and Au(111) with Au–I distance 2.95 and 2.86 Å. The C–*I* bond is more elongated on the Au(111) compared to Au(100) (2.17 Å vs. 2.13 Å) [26]; the C–I length of the free IB is 2.09 Å. Furthermore, the phenyl group is located on an Au atom. The iodine atom is strongly chemisorbed on the bridging and hollow sites. Regard to the activation of alkyne, the Ph–C≡C– fragment is also adsorbed on bridging and 3-fold hollow sites of Au(100) and Au(111). The coordination number of one Ph–C≡C– unit is 3 and 4 for Au(100) and Au(111) facets during the cross-coupling reaction, respectively, Figure 1, and the activation energy of cross-coupling is comparable [26].

Next, the activation of Ph−B(OH)_2_, Ph−I, and C≡C−H also was explored on a model cluster (Au_38_ and partially oxidized Au_38_O_2_) [30,31,32]. It is worthy to note that the model clusters of the Au_38_ and Au_38_O_2_ are somewhat different from the real catalyst in terms of the real structure. The Au_38_O_2_ cluster contains metallic Au^0^ and cationic Au^δ+^ species, and each O atom bonds to three Au atoms. In the activation process of phenylboronate, DFT showed that the phenylboronate reactant is preferentially adsorbed on the Au^0^ species rather than on the Au^δ+^ sites [30], leading to formation of the final product- biphenyl (Figure 2B). Meanwhile, DFT simulation also found that the interaction between the Au^δ+^ sites and Ph−I is weaker and the adsorption of C≡C−H on Au^δ+^ sites is relatively strong. Further, the proton of alkyne was detached with aid of O atom in the deprotonation step (Figure 2A). Therefore, a very low activation energy is required when the cross-coupling reactions occurred over the Au_38_O_2_ cluster. Of note, the cross-coupling step of the activated PA and –Ph on the Au_38_O_2_ should be the rate-determining step [32].

## 3. Physical Property of Au Nanoclusters

### 3.1. Framework

The gold clusters with crystal structure can be employed as the practical simulation models for mechanism study. In this Review, we only focus on the two cluster structures of the nanorod-shaped [Au_25_(PPh_3_)_10_L_5_Cl_2_]^2+^ (L = −SR and PA) and the nanosphere [Au_25_(SR)_18_]^x^ (*x* = −1, 0, +1, etc.), used as the real model for DFT studies (*vide infra*). The Au_25_(SR)_18_ cluster comprises an Au_13_ core [33] and six staples of Au_2_(SR)_3_ [34]. [Au_25_(PPh_3_)_10_L_5_Cl_2_]^2+^ is composed of two Au_13_ cores by sharing one common vertex to form the waist sites [35], connected by thiolate or alkyne ligands [36] (Figure 3).

### 3.2. Redox Property of Au cluster

Au_38_S_2_(SAdm)_20_ (SAdm = adamantanethiolate) nanoclusters exerted photosensitizing properties to give singlet oxygen (^1^O_2_) under visible light irradiation (e.g., 532 and 650 nm) [37]. The Au_38_S_2_(SAdm)_20_ is intact during the whole photocatalysis process, evidenced by UV-vis tracing and mass spectroscopy analysis. The cyclic voltammetry analysis showed that the Au_38_S_2_(SAdm)_20_ cluster had good charge transfer capacity to the redox K_3_Fe(CN)_6_ probe, Figure 4 [38]. However, the redox property of the cluster disappeared when *β*-cyclodextrins (*β*-CDs) was introduced in the THF solution. After detailed analysis, the huge *β*-CDs “umbrella” can trap the adamantane groups and then completely cover windows of the Au_38_S_2_(SAdm)_20_ nanoclusters, thereby blocking direct interaction with foreign molecules and then quenching the charge transfer process (Figure 4B). It indicated that these gold nanoclusters have good redox properties and electron transfer (ET) capacity during the catalytic reactions [39].

Further, Kumar and coworkers studied the Au_25_(SG)_18_ catalyst in an electrochemical oxidation [40]. The Au_25_(SG)_18_ on the electrode gave good electro-activity during the oxidation of ascorbic acid and dopamine over a wide linear range from 0.71 to 44.4 µM. And pH dependent electrocatalytic activity was observed, attributed to the consequence of pH-dependent electrostatic attraction/repulsion between the charged Au_25_(SG)_18_ clusters and the charged analytes. Moreover, an amperometric sensing method for other compounds was developed. Next, Kauffman et al. investigated the electron transfer between CO_2_ and Au_25_(PET)_18_ in solution [41]. Upon the DMF solution (containing Au_25_(SR)_18_) was saturated with CO_2_ gas, the optical absorbance features showed the oxidized state of Au_25_(SR)_18_. Meanwhile, the photoluminescence increases and blue-shift. The CO_2_^−^ induced optical changes can be simply reversed by purging the solution with N_2_ gas to remove the CO_2_, indicating an interaction between Au_25_(SR)_18_ and CO_2_. DFT calculations revealed that the CO_2_ molecule interacts with three S atoms of the Au_3_ site of the Au_25_(SR)_18_ cluster, prompting the CO_2_ electrochemical reduction. These observed unique interactions and electron transfers between gold clusters and reactants make the clusters promising catalysts during the organic transformations.

## 4. Catalytic Properties

### 4.1. Ullmann Coupling

At the beginning, the catalytic activity of the Au nanoclusters was examined in the Ullmann homo-coupling reactions of aryl iodides, which are generally catalyzed by palladium, nickel, and copper catalysts [42]. The supported gold cluster catalysts were simply prepared by a vortex-mixing of supports and a solution containing gold clusters at room temperature, and an annealing at 150 °C. The supported Au clusters were intact after the 150 °C annealing process (higher than the reaction temperatures), evidenced by UV−vis and scanning transmission electron microscopy (STEM) [43]. The X-ray photoelectron spectroscopy (XPS) analysis shows the chemical state of Au species in the oxide-supported cluster catalysts is positively charged (Au^δ+^) [44], where 0 < δ < 1, consistent with the free gold nanoclusters. The catalytic processes were carried out at 130 °C in the presence of base, which is similar with these catalyzed by Pd/Cu complexes or nanoparticles. The Au_25_(SR)_18_/CeO_2_ showed the best catalytic activity, and the test was then expanded to a serial of substituents with functional side-groups (Table 1) [45]. Of note, the efficiency of gold nanoclusters was not as good as the palladium, nickel, and copper nanocomposites in the Ullmann homo-coupling reactions of aryl chlorides and aryl bromides.

Later, these Au cluster were studied in the Ullmann hetero-coupling reactions. The catalytic conditions over the Au_25_(SR)_18_/CeO_2_ catalysts were the same with the homo-coupling reactions (Table 1 vs. Table 2). The aromatic and aliphatic thiolate-capped Au_25_ nanoclusters (e.g., naphthalenethiolate (-SNap), benzenethiolate (-SPh), hexanethiolate (-SC_6_H_13_), and 2-phenylethanethiolate (PET)) were chosen for comparison and exploration in the Ullmann hetero-coupling of 4-MeC_6_H_4_I and 4-NO_2_C_6_H_4_I [46]. Intriguingly, the aromatic thiolate ligated Au_25_ clusters gave much better catalytic performance (both the conversion of NO_2_C_6_H_4_I and selectivity for the hetero-coupling product (4-methyl-4′-nitrobiphenyl) than these protected by alkyl thiolate ligands. The Au_25_(SNap)_18_ cluster gave an 82% selectivity toward the hetero-coupling product, which was much higher than the Cu, Pd, and Au complexes (the selectivity: <30%, Table 2). Unfortunately, both of the conversion and selectivity decreased in the 2nd and 3rd cycles, which was due to the removal of the capping surface ligands and hence the decomposition of Au clusters, evidenced by the TEM images. These large gold nanoparticles jeopardized the catalytic performance in this coupling reaction. Thus, the protecting ligands on the clusters’ surface play a key influence on their catalytic properties.

DFT simulations were applied to explain the catalytic results. It is worthy to note that the reactants of both 4-MeC_6_H_4_I and 4-NO_2_C_6_H_4_I cannot interact well with the intact Au_25_(SR)_18_ clusters, because of the steric effect of the protecting thiolate ligands on the clusters’ surface. In the first step, one “-SR” unit on the Au_25_(SR)_18_ cluster was surmised to be detached under the reaction conditions in the presence of a K_2_CO_3_ base. Then the gold atoms on the motif were exposed to reactants and were associated with the catalytic sites [44]. Further, the activation energy for the homo- and hetero-couplings over the Au_25_ protected by “-SCH_3_” thiolate were compared by the nudged elastic band (NEB) approach (Figure 5). Intriguingly, the activation energy in the hetero-coupling was less than in the homo-coupling in the case of Au_25−_SNap clusters, (Figure 4). It implied that the aromatic thiolate-capped gold cluster can not only improve the conversion rate but can also favor the hetero-coupled process [46].

DFT simulations were applied to explain the catalytic results. It is worthy to note that the reactants of both 4-MeC_6_H_4_I and 4-NO_2_C_6_H_4_I cannot interact well with the intact Au_25_(SR)_18_ clusters, because of the steric effect of the protecting thiolate ligands on the clusters’ surface. In the first step, one “-SR” unit on the Au_25_(SR)_18_ cluster was surmised to be detached under the reaction conditions in the presence of K_2_CO_3_ base. Then the gold atoms on the motif were exposed to reactants and were associated with the catalytic sites [44]. Further, the activation energy for the homo- and hetero-couplings over the Au_25_ protected by “-SCH_3_” thiolate are comparable by the nudged elastic band (NEB) approach (Figure 5). Intriguingly, the activation energy in the hetero-coupling less than in the homo-coupling in the case of Au_25−_SNap clusters, (Figure 4). It implied that the aromatic thiolate-capped gold cluster not only can improve the conversion rate but also can favor the hetero-coupled process [46].

### 4.2. Suzuki Coupling

Further, the titania-supported Au_25_ clusters are studied in the Suzuki coupling in the presence of ionic liquids (ILs), which are catalyzed over palladium catalysts [42]. The Suzuki cross-coupling run at 90 °C using different solvents (e.g., ethanol, xylene, toluene, *N*,*N*′-dimethylformamide (DMF), ILs, etc.). The imidazolium-based ILs exerted a large effect on the MeOC_6_H_4_I conversion to the desire products. A very low conversion (<5%) is observed when using the ethanol, toluene, o-xylene, and DMF as solvents in the Au_25_/TiO_2_ catalyzed coupling reactions (Table 3). Interestingly, the iodoanisole conversion over Au_25_/TiO_2_ drastically increased to 89%–99% when BMIM·X (BMIM: 1-butyl-3-methylimidazolium, X = Br or Cl or BF_4_) solvents are introduced to the reaction system (Table 3). The catalytic results indicate that the imidazolium-based ILs acts as a promoter for the cross-coupling reactions [47]. Of note, only the BMIM cation (i.e., the acidic proton at position 2 of the imidazolium ions) play an important role during the reactions, as no activity is found in the presence of BDiMIM·BF_4_ (BDiMIM: 1-butyl-2,3-dimethylimidazolium) solvent, which is further supported by the DFT calculations. It is worthy to note that the efficiency of gold nanoclusters was not as good as the palladium nanocomposites, however, the gold nanoclusters exhibited much better selectivity toward the target cross-coupling products.

To explore the active species during the coupling reactions, the free Au_25_(PET)_18_ was mixed with the BMIM·BF_4_ under the same reaction conditions [43]. Except the molecular peak of Au_25_(PET)_18_ cluster, four new mass peaks are clearly detected in the matrix-assisted laser desorption/ionization mass spectrometry (MALDI-MS). These new appeared mass peaks belonged to the Au_25−*n*_(SR)_18−*n*_ (where, *n* = 1–4) species (Figure 6). Of note, these new species are not the fragments caused by laser of the MALDI method. These species also were observed in the ESI-MS method [48]. The imidazolium-based ILs indeed assist the yield of Au_25−*n*_(SR)_18−*n*_ species under the reaction conditions, which may be the active sites for the cross-coupling reactions. The other explanation is that the Au-NHC complex (NHC: N-heterocyclic carbene) with the Au_25−*n*_(SR)_18−*n*_ species could be responsible for the active sites during the Suzuki cross-coupling reactions, although it needs further investigation. Of note, the Au-NHC complex was the product of the reaction of BMIM cations with the gold nanoclusters.

### 4.3. Sonogashira Coupling

As the IB and alkyne can be activated over gold clusters, hence, the catalytic performance of the gold nanoclusters may extend to Sonogashira cross-coupling reactions, often catalyzed over palladium catalysts [42]. The catalytic performance of the Au_25_(PET)_18_ cluster (supported on oxides) in the Sonogashira cross-coupling reaction was studied [49]. The supported catalyst was prepared by impregnating oxide powders (such as TiO_2_, CeO_2_, SiO_2_, and MgO) in a CH_2_Cl_2_ solution of Au_25_(PET)_18_ (~1 wt % loading) with a 150 °C annealing. STEM and TG analyses showed that the protecting thiolate ligands were intact on the surface of gold clusters after thermal treatment. Then these Au_25_/oxide catalysts were applied to the Sonogashira cross-coupling reaction. The optimized reaction conditions were using DMF as a solvent and K_2_CO_3_ as a base under an N_2_ atmosphere at 160 °C, which is harsher than those for the above Suzuki and Ullmann couplings. The Au_25_/CeO_2_ catalyst showed the best activity (96.1% iodoanisole conversion with 88.1% selectivity toward the target product) (Table 4). The solvent and base can also influence the product selectivity. The size-dependent catalytic performance also was studied.

The catalytic performance of small-sized Au_25_(PET)_18_ cluster catalysts was much better than large-sized of gold clusters of 2–3 nm and Au/CeO_2_ (~20 nm). Support effects were studied in the cross-coupling, and no distinct effect of the oxide supports was observed (i.e., CeO_2_, SiO_2_, TiO_2_, and MgO). The conversion was no obvious decrease, but the selectivity decreased from 88.1% to 64.5% after 5 cycles. It is noteworthy that TEM analysis shows that the gold clusters grow into larger nanoparticles (>3 nm), meaning that the gold clusters capped by organic ligands cannot stay intact under harsh reaction conditions (160 °C in the presence of a base). The gradual degradation of gold clusters leads to a decrease in selectivity, as the larger Au clusters showed a much lower selectivity. It is worthy to note that the efficiency of gold nanoclusters is much worse than the palladium-based catalysts, and the selectivity for the cross-coupling products over Au clusters is also worse.

DFT calculation found that the reactants (i.e., IB and PA) prefer to adsorb on the open facet (Au_3_) of the Au_25_ cluster with the phenyl ring facing a surface Au atom (Figure 7). A total adsorption energy reaches −0.90 eV when the two reactants co-adsorb on the Au_25_(SR)_18_ catalyst. While, the IB/IB pair has an adsorption energy of −1.05 eV, indicating that the IB/IB pair interacts strongly with the cluster and the homocoupling of IBs is the dominant side-reaction competing with the cross-coupling between IB and PA. DFT results suggested that the catalytic active sites is associated with the Au_25_(SR)_18_ clusters, which is consistent with the experimental results.

The structure of the 25-atom cluster is similar [50], but the electronic property and the catalytic activity of the bimetallic clusters can be largely regulated by the foreign dopants [50,51,52,53,54,55]. Recently, Li et al. [56] studied the doping effects of the Au_25_(SR)_18_ nanoclusters in the Sonogashira cross-coupling reaction base on the experiment and DFT simulations. The obtained results suggested that the Cu and Ag atoms are preferentially occupied at the cluster’s kernel (Au_13_) rather than the Au_2_(SR)_3_ staple motif, while a single Pt atom only can be doped individually and locates in the center of the cluster. The overall performance of Ag*_x_*Au_25−*x*_(SR)_18_ was similar to that of Au_25_(SR)_18_ and Pt_1_Au_24_(SR)_18_, which showed a decrease in catalytic activity (Table 5). The catalytic activity was from Ag*_x_*Au_25−*x*_(SR)_18_ ≈ Au_25_(SR)_18_ > Cu*_x_*Au_25−*x*_(SR)_18_ > Pt_1_Au_24_(SR)_18_. Interestingly, the Cu*_x_*Au_25−*x*_(SR)_18_ produced a homo-coupling product base on the Ullmann homo-coupling pathway, which is contrary to the other three cluster catalysts. However, DFT calculations showed that the adsorption energy of one PA molecule on the Pt_1_Au_24_(SR)_18_, Cu_1/2_Au_24/23_(SR)_18_, and Au_25_(SR)_18_ nanoclusters was very similar (−0.50 to −0.52 eV, Table 6). The adsorption energy of one IB molecule onto the Pt_1_Au_24_(SR)_18_, Ag_1/2_Au_24/23_(SR)_18_ and Au_25_(SR)_18_ was also very similar (−0.59 to −0.61 eV, Table 6). These results suggested that the adsorption process of the PA and IB onto the alloy clusters is not the key step during the coupling reactions. Generally, the catalytic activity is largely affected by the electronic effect in the core of bimetallic clusters (i.e., Pt_1_Au_12_, Cu*_x_*Au_13-*x*_, Ag*_x_*Au_13-*x*_, and Au_13_), and the selectivity of product is primarily turned by the atomic type on the shell of M*_x_*Au_12-*x*_ [51,57].

The structure of the 25-atom cluster was similar [50], but the electronic property and the catalytic activity of the bimetallic clusters could be largely regulated by the foreign dopants [50,51,52,53,54,55]. Recently, Li et al. [56] studied the doping effects of the Au_25_(SR)_18_ nanoclusters in a Sonogashira cross-coupling reaction based on an experiment and DFT simulations. The obtained results suggested that the Cu and Ag atoms were preferentially occupied at the cluster’s kernel (Au_13_) rather than the Au_2_(SR)_3_ staple motif, while a single Pt atom only can be doped individually and locates in the center of the cluster. The overall performance of Ag*_x_*Au_25−*x*_(SR)_18_ was similar to that of Au_25_(SR)_18_ and Pt_1_Au_24_(SR)_18_, which showed a decrease in catalytic activity (Table 5). The catalytic activity was Ag*_x_*Au_25−*x*_(SR)_18_ ≈ Au_25_(SR)_18_ > Cu*_x_*Au_25−*x*_(SR)_18_ > Pt_1_Au_24_(SR)_18_. Interestingly, the Cu*_x_*Au_25−*x*_(SR)_18_ produced a homo-coupling product base on the Ullmann homo-coupling pathway, which was contrary to the other three cluster catalysts. However, DFT calculations showed that the adsorption energy of one PA molecule on the Pt_1_Au_24_(SR)_18_, Cu_1/2_Au_24/23_(SR)_18_, and Au_25_(SR)_18_ nanoclusters was very similar (−0.50 to −0.52 eV, Table 6). The adsorption energy of one IB molecule onto the Pt_1_Au_24_(SR)_18_, Ag_1/2_Au_24/23_(SR)_18_, and Au_25_(SR)_18_ was also very similar (−0.59 to −0.61 eV, Table 6). These results suggested that the adsorption process of the PA and IB onto the alloy clusters was not the key step during the coupling reactions. Generally, the catalytic activity was largely affected by the electronic effect in the core of bimetallic clusters (i.e., Pt_1_Au_12_, Cu*_x_*Au_13-*x*_, Ag*_x_*Au_13-*x*_, and Au_13_), and the selectivity of the product is primarily turned by the atomic type on the shell of M*_x_*Au_12-*x*_ [51,57].

### 4.4. A^3^-Coupling

A^3^-coupling reactions, where three reactants (aldehydes, amines, and alkynes) react each other in one-pot to yield only one product, have attracted overwhelming interest in the past decade. The A^3^-coupling is favorable from environmental and economic perspectives: more efficient and less waste [58,59,60,61]. The A^3^-coupling reactions involve new C–C and C–N bond formation in one procedure. The alkyne activation was deemed as the key step for the A^3^-coupling; the aldehydes could react with amines spontaneously. Hence, the gold clusters should be active in this reaction, because the cluster catalyst has exhibited the capacity of the alkyne activation in the semi-hydrogenation of terminal alkynes [36].

The catalytic performance of the Au_25_(PPh_3_)_10_(PA)_5_X_2_ was investigated in different solvents and reaction temperatures [62]. The most prominent feature was that the polarity of the solvent had a great influence on catalytic activity. The TiO_2_-supported Au_25_ catalyst gave higher activity in the polar solvents. The gold clusters also showed good recyclability. Interestingly, the cluster catalyst showed no conversion using ketones as reactants (Figure 8), which was completely different from the catalytic behaviors of the gold complexes and bare gold nanoparticles. Therefore, the electronic factors and steric hindrance of the substituents had significant effects on the reaction conversion rate.

An induction period (0 to 3 h) appeared and the conversion slightly increased during the induction period. After the induction period, the reaction conversion of gold clusters significantly increased in the time evolution for A^3^-coupling [62]. The results showed that some phosphine ligands are removed to generate catalytic active sites, which associates with the surface gold atoms. Further research found that the capped phosphine ligands can be selectively removed in the case of Au_11_(PPh_3_)_7_X_3_ with the aid of base (e.g., pyridine), evidenced by UV-vis and ESI-MS analyses [63].

Finally, the catalytic mechanism over the gold clusters was studied by DFT calculations. Firstly, the phosphine ligand is detached in the presence of the reaction system, and then the uncovered Au atoms are exposed to the reactants. Next, the PA molecules are adsorbed onto the M1 site via the interaction of Au···whole triple bond, Figure 9. Further, a terminal hydrogen deprotonation occurrs in the presence of amine (e.g., HN(CH_3_)_2_). The iminium ion (H_2_C=N(CH_3_)_2_^+^) interacts with the PhC≡C– on site M1 and finally give rise to the final product (i.e., propargylamine) [62].

Further, Jin et al. reported Au_38_ nanocluster-catalyzed the A^3^-coupling reaction [64]. They argued that the synergistic effect of the partial positive charged Au surface (Au^δ+^, 0 < δ < 1) and the electron-rich Au_23_ kernel were responsible for the catalytic behaviors. Li et al. found that cadmium doped Au_13_ nanocluster also showed catalytic activity in the A^3^-coupling [65]. The cooperation between the exerted cadmium atoms and the neighbor gold atoms on the surface of Au_13_ icosahedron are tentatively deemed as the active sites in the cross-coupling reactions.

## 5. Summary and Outlook

In the past decade years, remarkable advances have been made in developing catalytic applications of gold nanoclusters, which is a new perspective for gold nanocatalysis, especially in carbon–carbon cross-coupling reactions. These gold nanoclusters were more efficient in the cross C-C coupling reactions in this Review. The higher catalytic activity was mainly because of the distinctive frame structure and electronic properties of the gold nanoclusters and the protecting ligands. However, some of the gold nanoclusters with the protection of thiolate/phosphine could not stay intact under the harsh reaction conditions during the catalysis processes, leading to a decrease or even disappearance of catalytic activity due to the increasing size of formed particles. How to maintain the stability of gold nanoclusters during the catalysis has become a major challenge and subject for future research.

The present review demonstrated that CeO_2_ and TiO_2_ oxides have been observed to be excellent supports for gold nanoclusters [66]. Few studies have been done on other oxides. Developing new types of supports is another significant issue for substances such as zeolite [67,68], carbon materials (e.g., graphene or graphite oxide) [69,70], and MOFs [71]. For example, mesoporous materials, e.g., MOFs and zeolite, exhibited regular tunnels and cages and improved the stability of the Au clusters during the catalytic processes [72]. The tunnels and cages, by enriching concentration of the reactants, can also improve the catalytic activity. The steric effects of MOFs organic linker can improve the selectivity of products. In addition, the sites of Lewis acids and Bronstein acids in zeolites also can change the catalytic performance of the catalysts.

In a word, gold nanoclusters will be expected to be used into carbon–heteroatom cross-coupling reactions, which are not yet well investigated. The formed C–O and C–N bonds are very useful drug intermediates [73]. Recently, some attempts have been reported in these categories, such as the cross-aldol condensation and Michael addition (forming C=C bonds) and photo-oxidation of amines to imines (C=N bonds) [10,74]. Future research on gold nanoclusters will contribute to fundamental researches and provide clues for the new design of efficient catalysts for other specific chemical processes.

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
