# Peer review of "Heterogeneous Cross-Coupling over Gold Nanoclusters"

_nanomaterials, 2019, doi:10.3390/nano9060838_

Round 1
Reviewer 1 Report
This new revised version of the manuscript has been improved and could be accepted if the following points are addressed:
-Table 1, "conversion" not "coversion"
-Table 1: in Au25 (once in the title and above the arrow), the underscript 25 should not be in italics.
-Table 2, K2CO3 not k2CO3
-The title of section 3.2 is misleading. Furthermore, the cited example is not the only one present in the literature pointing out the ET capabilities of these Au25 nanoclusters. If the authors want to maintain this section, they should carefully check the literature and update it.
Author Response
This new revised version of the manuscript has been improved and could be accepted if the following points are addressed:
A: Thanks for the comments to improve our paper.
-Table 1, "conversion" not "coversion"
A: It is done.
-Table 1: in Au25 (once in the title and above the arrow), the underscript 25 should not be in italics.
A: It is done.
-Table 2, K2CO3 not k2CO3
A: It is done.
-The title of section 3.2 is misleading. Furthermore, the cited example is not the only one present in the literature pointing out the ET capabilities of these Au25 nanoclusters. If the authors want to maintain this section, they should carefully check the literature and update it.
A: Thanks for the suggestion. We change the title of section 3.2 to “Redox Property of Au cluster”
Reviewer 2 Report
The research article “Heterogeneous Cross-Coupling over Gold Nanoclusters" is well written. But the authors need to add information before the publication. Thus a major revision is suggested.
Comments
Authors compared functionalization of other nanomaterials with Au clusters for the catalytic performance. At the same time, authors should compare the other nanocomposites with the Au clusters composite for catalytic reaction in the table form.
How gold can play an efficient catalyst than other nanocomposites in cost effective manner?
From the headings 2-4, authors should discuss about more reference papers. Almost the author referred the same reference for the whole sub-headings. The references are repeated and reference are not maintained properly.
The reference 48 is not refer to ‘25-atom cluster is similar’. Authors should include the proper reference.
Author Response
Authors compared functionalization of other nanomaterials with Au clusters for the catalytic performance. At the same time, authors should compare the other nanocomposites with the Au clusters composite for catalytic reaction in the table form. How gold can play an efficient catalyst than other nanocomposites in cost effective manner?
A: Many thanks for Reviewer’s suggestion. We well agree with the Reviewer’s point; the comparison of other nanocomposites with the Au clusters is very important. We add some contents in the revised manuscript.
From the headings 2-4, authors should discuss about more reference papers. Almost the author referred the same reference for the whole sub-headings. The references are repeated and reference are not maintained properly.
A: Thanks for Reviewer’s suggestion. We discuss more reference papers in the revised manuscript.
The reference 48 is not refer to ‘25-atom cluster is similar’. Authors should include the proper reference.
A: We changed the right reference.
Round 2
Reviewer 1 Report
The authors changed the manuscript according to the suggestions.
In my previous report, I suggest to change the title of Section 3.2 and to
carefully check the literature, expand and update it; the authors change the title but they do not change the literature regarding ET with these type of clusters.
Author Response
In my previous report, I suggest to change the title of Section 3.2 and to carefully check the literature, expand and update it; the authors change the title but they do not change the literature regarding ET with these type of clusters.
A: Many thanks for the suggestion. We have expanded the Section 3.2 and some new literatures on the ET are added in the revised manuscript.

Reviewer 2 Report
authors changed the manuscript according to the reviewer's comments. The manuscript is well organized and well written.
Author Response
authors changed the manuscript according to the reviewer's comments. The manuscript is well organized and well written.
A: Many thanks for the comments.
Round 3
Reviewer 1 Report
The manuscript could be accepted in the present form. The references cited in section 3.2 remain still limited and very specific as examples. Since this section describes the properties of the Au25 clusters, when I asked to add more references and comments I was thinking on manuscripts describing mainly the ET process on the clusters or the CV of the clusters alone.